# Association Between the Healthy Eating Index and the Body Mass Index of Older Adults: An Analysis of Food Frequency and Preferences

**DOI:** 10.3390/nu17101717

**Published:** 2025-05-19

**Authors:** Andres Fontalba-Navas, Ruth Echeverria, Cristina Larrea-Killinger, Mabel Gracia-Arnaiz, Claudia Soar, Juan Pedro Arrebola

**Affiliations:** 1Antequera Hospital, Northern Málaga Integrated Healthcare Area, 29200 Antequera, Spain; 2Department of Public Health and Psychiatry, University of Málaga, 29010 Málaga, Spain; 3Group BE-21, Institute of Biomedical Research in Málaga (IBIMA Plataforma BIONAND), 29010 Málaga, Spain; 4Fundación para la Investigación Biomédica del Hospital Universitario 12 de Octubre (FIBH12O), Instituto de Investigación Sanitaria Hospital 12 de Octubre (imas12), 28041 Madrid, Spain; recheverria.imas12@h12o.es; 5“Toxic Body” Interdisciplinary Network, Department of Social Anthropology, University of Barcelona, 08001 Barcelona, Spain; larrea@ub.edu; 6Department of Social Anthropology, University of Barcelona, 08001 Barcelona, Spain; 7Research Group Anthropology of Crisis and Contemporary Transformations (CRITS), Department of Social Anthropology, University of Barcelona, 08001 Barcelona, Spain; 8INSA-UB (Nutrition and Food Safety Research Institute) María de Maeztu Unit of Excellence, University of Barcelona, 08921 Santa Coloma de Gramanet, Spain; 9CIBER de Epidemiología y Salud Pública (CIBERESP), 28029 Madrid, Spain; 10Department of Anthropology, Philosophy and Social Work, Universitat Rovira i Virgili, 43002 Tarragona, Spain; mabel.gracia@urv.cat; 11Nutrition Post-Graduate Program, Department of Nutrition, Federal University of Santa Catarina, Florianopolis 88040-900, Brazil; claudia.soar@ufsc.br; 12Department of Preventive Medicine and Public Health, Pharmacy School, Universidad de Granada, Campus de Cartuja s/n, 18071 Granada, Spain; 13Instituto de Investigación Biosanitaria (ibs.GRANADA), Avda. de Madrid, 15. Pabellón de Consultas Externas 2, 2a Planta, 18012 Granada, Spain; 14Consortium for Biomedical Research in Epidemiology and Public Health (CIBERESP), Instituto de Salud Carlos III, C/ Monforte de Lemos 3-5, Pabellón 11. Planta 0, 28029 Madrid, Spain

**Keywords:** Healthy Eating Index, Body Mass Index, older adults, food security, obesity, ultra-processed foods

## Abstract

**Background/Objectives:** The nutritional habits of older adults are increasingly relevant to public health, particularly given the rising prevalence of obesity and its associated chronic diseases. This study aims to analyze the relationship between the Healthy Eating Index (IASE) and Body Mass Index (BMI) in older adults in Spain, focusing on food frequency, dietary preferences, and socioeconomic factors influencing nutritional security. **Methods:** The study is part of the Eating Matters project, assessing food (in)security in older adults across Andalusia and Catalonia between April 2022 and January 2024. A cross-sectional survey was conducted among 190 participants (≥65 years), recruited in primary healthcare centers. The questionnaire included three blocks: food insecurity assessment (FIES scale), diet quality with the Healthy Eating Index for the Spanish Population (IASE), and sociodemographic factors. Data analysis involved descriptive statistics, Pearson correlations, and logistic regression models to identify associated factors with overweight and obesity. **Results:** The average BMI was 28.5 kg/m^2^ (SD = 4.29), with 46.3% of participants classified as overweight and 32.1% as obese. A significant negative correlation (r = −0.79, *p* < 0.05) was found between healthy food consumption and BMI, while personal income showed a moderate positive correlation with adherence to a healthy diet (r = 0.42, *p* < 0.05). Logistic regression indicated that frequent consumption of processed meats and confectionery was a significant identify associated factors with overweight/obesity, with a model accuracy of 68% and sensitivity of 95%. **Conclusions:** Older adults with lower incomes and higher consumption of ultra-processed foods exhibited a higher risk of obesity. These findings highlight the need for public policies promoting food accessibility and targeted nutrition education for older adults, including guidance on balanced diets, adequate protein intake, and the prevention of sarcopenia, to encourage healthier dietary patterns in aging populations.

## 1. Introduction

In Spain, the nutrition of older adults is an issue of growing research interest, given the progressive aging of the population and the challenges this poses in terms of public health [1]. With rising life expectancy, which by 2022 had reached 80.7 years in the European Union and 83.2 years in Spain [2], it is essential to pay close attention to the eating habits of the elderly, and to devise and apply public policies to promote safe, fair, adequate, healthy, and sustainable nutrition, in view of its importance in preventing chronic diseases and improving the quality of life.

People in this age group are subject to overweight and obesity. The phenomenon of increasing obesity among older people is not exclusive to Spain, but has been observed worldwide, as demonstrated by various population studies. The problem affects both men and women, although it is observed more frequently in the latter [3,4,5], and is present in many regions, spurred by demographic, socioeconomic, and lifestyle factors, further complicating the health status of older adults.

Numerous studies have recorded a strong association between obesity and the consumption of ultra-processed, high-calorie foods (UPFs) [6,7]. These products, which often contain high levels of added sugars, saturated fats, and sodium and lack essential nutrients, are designed to be highly palatable and appetizing, thus stimulating overconsumption. Moreover, UPFs tend to displace healthier options, contributing to high caloric intake without providing a corresponding nutritional input. High consumption of UPFs by older people can also aggravate health problems such as type 2 diabetes, cardiovascular disease, and other metabolic disorders that are closely linked to obesity [8,9].

Vulnerable older people, especially those with economic, physical, or social limitations, may be predisposed to consume UPFs for several reasons. First, these foods are usually more accessible and affordable than fresh alternatives, making them attractive to people with limited means. Furthermore, the ease of preparation of these products is of great importance for people with a physical or cognitive impairment, and also for older people who live alone, since cooking fresh foods can be more difficult and laborious. In addition, the mass distribution of UPFs and their ready availability in local stores facilitate their consumption by older adults, especially in areas with poor accessibility to fresh foods [9].

Among the UPFs most commonly consumed in Spain are confectionery and processed meats. The latter, such as sausages, chorizo, and cooked ham, are industrial foods that contain additives, preservatives, and large amounts of added salt and fats. Baker’s confectionery products (such as biscuits, pastries, and cakes) include refined flours, added sugars, and hydrogenated vegetable oils. They are easily accessible, low-cost foods, which are often consumed by older people with dental problems or difficulties in chewing [10]. These factors, together with the decreased sense of taste often experienced in later age, make UPFs such as processed meats and confectionery very popular, with markedly negative consequences for the health of vulnerable older people [11,12].

In the present article, we analyze the dietary patterns of older adults in Spain, their relationship with overweight/obesity, and the resulting implications for public health and wellbeing among this population group. Our research goal is to provide evidence on key determinants of food security and nutritional habits in an elderly population, a group traditionally regarded as vulnerable to health-related factors. By leveraging the findings of this study, we seek to inform policies and health promotion programs, thus improving wellbeing and reducing the risks of chronic disease among this population, whilst promoting sustainability and fairness in the food system.

## 2. Materials and Methods

### 2.1. Study Goals: The ‘Eating Matters’ Project

The main objective of this study was to conduct a detailed survey of food (in)security in people aged over 65 years in the Spanish autonomous communities of Andalusia and Catalonia, between April 2022 and January 2024.

This research is part of a larger project, called Eating Matters, whose main goal is to investigate food (in)security problems affecting older people in Spain, via qualitative and quantitative methods applied to the population over 65 years of age in rural and urban areas of Andalusia, Catalonia, and the Valencian Community in the Spanish Mediterranean area [13,14,15]. The study has been approved by the Ethics Committees of the University of Barcelona, the Rovira i Virgili University, the Jordi Gol Foundation in Catalonia, and Northern Malaga Healthcare Area in Andalusia, corroborating its compliance with ethical standards and confidentiality requirements.

### 2.2. Recruitment of Participants and Sample Size

Participants were recruited from primary healthcare centers in Andalusia and Catalonia, specifically in the provinces of Málaga, Tarragona, and Barcelona. These centers provided an ideal setting due to their accessibility and the high level of trust patients place in healthcare staff, which helped minimize response bias. Eligible participants were individuals aged 65 years or older, residing in Spain for at least 40 years, and capable of providing informed consent. Those with severe cognitive impairments, language barriers, or who declined participation were excluded. Recruitment followed a randomized selection process using predefined lists at each site to ensure a representative sample across geographic and sociodemographic profiles. The required sample size was calculated assuming a food insecurity prevalence in vulnerable older adults, with a 95% confidence level and a 6% margin of error. A minimum of 171 participants was needed. To account for potential non-response, the final target was increased to 200. This strategy ensured sufficient statistical power and robustness for data analysis [16].

### 2.3. Questionnaire Design

The questionnaire was designed to address key aspects of nutrition in older people, and is divided into three distinct blocks:Block 1: Assessment of food insecurity using the Food Insecurity Experience Scale (FIES), an internationally validated tool, composed of eight questions [17];Block 2: Assessment of diet quality was conducted using the Healthy Eating Index for the Spanish Population (IASE, Spanish initials). This validated instrument includes nine items that measure the frequency of consumption of key food groups and is aligned with national dietary recommendations. The SCORE IASE refers to the total score obtained from the IASE, which quantifies adherence to healthy dietary patterns—higher scores indicate better diet quality. The IASE was selected for this study as it provides a straightforward and culturally appropriate assessment of diet quality within the Spanish context, and it has been previously used and validated in populations aged over 65 years [18,19,20];Block 3: Sociodemographic data, medical history, diet and exposure to contaminants in food and food contact materials, sources of food supply, and perceptions of food (in)security. This block, exploratory in nature, was internally validated by a group of experts after implementing a pilot survey with 10 participants, a process which enabled any adjustments in the wording and sequence of the questions that might be necessary.

To ensure the validity and applicability of the questionnaire, a pilot study was conducted with 10 participants aged 65 years and older. The pilot tested content and face validity, evaluating the clarity, relevance, and understanding of all items. Based on the feedback received, minor modifications were made to improve question wording and sequencing. This process ensured that the final questionnaire was suitable for the target population and facilitated accurate data collection.

In older adults, the interpretation of Body Mass Index (BMI) differs from that of the general adult population due to age-related physiological changes such as reduced muscle mass and altered fat distribution. Evidence suggests that a BMI between 22 and 27 kg/m^2^ is associated with better outcomes and lower mortality in the elderly. Therefore, the classification in this population is typically as follows: underweight is defined as a BMI less than 22 kg/m^2^, normal weight ranges from 22 to 27 kg/m^2^, overweight is considered between 27 and 30 kg/m^2^, and obesity is defined as a BMI greater than 30 kg/m^2^. These thresholds are recommended by geriatric and nutritional societies such as the European Society for Clinical Nutrition and Metabolism (ESPEN) and are supported by research showing that slightly higher BMI values in older individuals may have a protective effect against frailty and mortality [21]. Income data included individual and household monthly earnings from all sources, such as pensions, rental income, and investment returns, as reported by the participants.

### 2.4. Data Collection and Quality

The study data were collected in a quiet environment at the Health Centre, where participants’ comfort was assured, and explanations could be provided when necessary. The interviewers, who were staff members at the Health Centre, carried out regular quality control exercises to ensure the consistency and integrity of the data collected, and a member of the research team supervised the purging of the database.

To ensure patient confidentiality, each participant was pseudonymized using a unique code. Printed data were destroyed once transferred to the digital database, which was stored following open science protocols. This database will be deposited in the Social Science Database (ARCES, Spanish initials) managed by the Centre for Sociological Research, within the Ministry for the Presidency, Relations with Parliament and Democratic Memory, in accordance with open science and transparency requirements.

### 2.5. Data Analysis

When data collection was complete, a descriptive statistical analysis was made of each of the three questionnaire blocks. Of the 199 surveys returned, 9 did not meet the inclusion/exclusion criteria and were discarded. A preliminary analysis of the remaining 190 included the identification of trends and possible problems in data quality, such as the tendency of participants to give homogeneous answers to the questions in the third block. For future application of the survey, this potential problem was resolved by changing the order of the blocks.

Statistical analyses included bivariate tests (Chi-square and Fisher’s exact test) to explore the relationships between the variables of the three blocks, as well as multivariable logistic regression to identify associated factors with obesity. Given the skewed distribution of many of the variables, non-parametric approaches were applied to ensure the robustness of the statistical analyses. The aim of these analyses was to enable evidence-based recommendations for the design of public policies addressing the nutritional challenges of older people within a context of growing social inequality. All these analyses were performed using the R statistical software package (Version 4.4.1) [22].

## 3. Results

The sample population was composed of 190 participants with a mean age of 77.3 years (SD = 7.37). With respect to anthropometric characteristics, the average Body Mass Index (BMI) was 28.5 kg/m^2^ (SD = 4.29). A total of 46.3% of participants were classified as overweight, 32.1% as obese, and 21.6% as normal weight. The average weight was 74.5 kg (SD = 12.77) and the average height, 161.6 cm (SD = 8.14) (Table 1).

Responses to the ‘Healthy diet’ questionnaire showed that 123 participants (68.3%) believe their diet “needs changes”, acknowledging that their current eating habits and nutritional health are sub-optimal. Four persons (2.2%) indicated that their current diet was “unhealthy” and might be significantly contributing to the risk of overweight or obesity. By contrast, 53 participants (29.4%) claimed to have a “healthy” diet.

The relationship between BMI and the frequency of healthy food consumption was assessed using Spearman’s correlation coefficient. A significant negative correlation was observed (rho = –0.79, *p* < 0.05), indicating that individuals who consume healthy foods more frequently—such as fruits and vegetables—tend to have lower BMI values. This result suggests that a balanced diet rich in nutritious foods is associated with a reduced risk of overweight and obesity, emphasizing the importance of promoting consistent healthy eating patterns within this population.

The SCORE IASE is a validated index that measures adherence to the Mediterranean diet based on the frequency of consumption of key food groups such as fruits, vegetables, legumes, whole grains, fish, and olive oil, while limiting red meat and processed foods.

Our analyses also examined the relationship between personal income levels and the score obtained on the IASE healthy eating scale. Using the Spearman correlation coefficient, a moderate positive correlation was found between these variables (rho = 0.42, *p* < 0.05). This suggests that participants with higher incomes tend to obtain higher scores on the IASE scale, indicating greater access and adherence to healthy eating patterns. This result reinforces the idea that income plays an important role in the ability to maintain health-promoting eating habits, although other factors may also influence this relationship.

The association between the SCORE IASE and body mass index (BMI) was examined using Spearman’s rank correlation. The analysis revealed a statistically significant but weak negative correlation (rho = −0.089, *p* = 0.021), indicating that greater adherence to the Mediterranean diet was only modestly associated with lower BMI. Although statistically significant, the strength of this association is limited and should be interpreted with caution. Nevertheless, this finding suggests a potential inverse relationship between healthy dietary patterns and body weight status in the studied population.

Similarly, the relationship between individual income level and adherence to the Mediterranean diet was also assessed using Spearman’s rank correlation. The analysis showed a weak but statistically significant positive correlation (rho = 0.062, *p* = 0.039), suggesting that higher income levels were associated with better adherence to the Mediterranean dietary pattern

Table 2 shows the variables significantly associated with increased Body Mass Index (BMI), based on a multivariable logistic regression model. The analysis revealed a consistent relationship between lower dietary quality scores and higher odds of being overweight. Specifically, lower scores for vegetable consumption—which indicate a reduced intake of vegetables—were significantly associated with increased likelihood of being overweight (OR = 0.887; 95% CI: 0.790–0.997; *p* = 0.045). Likewise, lower scores on the confectionery and processed meats indices, which reflect greater consumption of sweets and processed meats, were also significantly associated with higher odds of overweight (SCORE.CONFECTIONERY: OR = 0.923; 95% CI: 0.843–0.999; *p* = 0.049; SCORE.PROCESSED MEATS: OR = 0.905; 95% CI: 0.825–0.993; *p* = 0.037).

These findings indicate that lower dietary quality scores—characterized by lower vegetable intake and higher consumption of confectionery and processed meats—are associated with increased BMI.

## 4. Discussion

The results of this study align with prior research indicating a direct association between unbalanced diets and the prevalence of obesity in older adults, particularly in Western societies where ultra-processed foods (UPFs) are widely accessible and often more affordable than healthier alternatives [7]. The high consumption of these foods among participants reflects earlier findings that identified processed meats (e.g., chorizo, ham, sausages) and bakery products (e.g., biscuits, pastries) as the primary UPFs consumed by older populations in these regions [10,12]. The low percentage of participants reporting a healthy diet underscores the urgent need for targeted interventions aimed at improving nutritional quality. Such strategies should not only focus on reducing UPF consumption but also emphasize the inclusion of fresh, nutrient-dense foods in older adults’ diets. Nutrition education campaigns tailored to this demographic, alongside public policies that enhance access to healthy foods, could substantially reduce obesity rates in this vulnerable group [23]. Our analysis revealed a strong inverse relationship between the frequency of healthy food consumption and BMI. Participants who reported regular intake of fruits and vegetables had notably lower BMI values compared to those with infrequent or no consumption. This underscores the pivotal role of a balanced diet in weight regulation and suggests that fostering healthier eating behaviors among older adults could be a key strategy in improving health outcomes and lowering the risk of chronic, diet-related conditions [24].

Conversely, low socioeconomic status appears to negatively affect dietary quality and, consequently, BMI. Individuals with limited financial resources often face barriers to accessing fresh and nutritious food, leading to reliance on cheaper, calorie-dense UPFs high in fat, sugar, and sodium. Economic hardship may also induce stress that influences food choices, driving consumption of comfort foods that are typically less nutritious—contributing to long-term metabolic health risks [25,26].

These findings highlight the necessity of implementing preventive nutritional policies that go beyond dietary education to address broader social determinants of health, including income, education level, and geographic accessibility to healthy food sources [27,28,29]. Moreover, lifelong dietary patterns significantly influence food choices in later life. Habits formed in early and middle adulthood often persist into older age, reinforcing both healthy and unhealthy behaviors. As such, preventive efforts should extend across the life course, promoting nutritional literacy from a young age to establish sustainable and health-supportive eating practices over time [30].

This study has some limitations. Although detailed information on income, education level, and household composition was collected, the present analysis focused specifically on dietary patterns and their association with body mass index. Future analyses will explore the influence of other sociodemographic variables. Income and education are often closely correlated and represent overlapping aspects of socioeconomic status. In this study, we focused on income as a more direct proxy for food accessibility, while acknowledging that the independent effect of education could not be fully isolated due to potential multicollinearity. Additionally, the cross-sectional design limits the ability to establish causal relationships. Longitudinal studies are needed to better understand the complex interplay between socioeconomic factors, dietary habits, and obesity in older adults. Another limitation of this study is the reliance on self-reported food questionnaires, which are inherently subject to recall bias and possible misreporting of dietary intake. While the instruments used (FIES and IASE) are internationally validated, their use depends on participants’ memory and subjective perception, which may affect data accuracy. Prospective methods or direct observation could provide more precise dietary information but were not feasible in this context.

Finally, social and familial contexts play a critical role in dietary adherence among older adults. Evidence suggests that strong social support networks are associated with better adherence to healthy diets, while social isolation correlates with higher risks of malnutrition and obesity [31,32]. Community-based programs that involve family members, caregivers, and local institutions have shown greater effectiveness than individual-focused interventions. When integrated with policies that subsidize fresh produce and regulate the availability of UPFs, such initiatives have the potential to yield broader and more sustainable public health benefits.

Overall, this study underscores the importance of a multidimensional and integrated approach to nutritional health in older adults—one that simultaneously addresses economic, educational, and environmental barriers to healthy eating.

## 5. Conclusions

The nutrition of older people is a major public health issue in Spain, which has an ageing population subject to increasing levels of obesity. According to our study results, factors such as economic limitations and the consumption of ultra-processed foods, like pastries, biscuits, and processed meats, contribute to unhealthy dietary patterns that increase the risk of obesity and hence chronic disease. These results underline the need for preventive public policies with a multidimensional approach aimed at promoting a more balanced and accessible diet, thus improving the quality of life and reducing the health risks associated with overweight and obesity in old age.

## Figures and Tables

**Table 1 nutrients-17-01717-t001:** Sociodemographic, economic, and BMI data.

	Andalusia		Catalonia		TOTAL	
Participants (*n*)	91	47.9%	99	52.1%	190	100.0%
Women	59	64.8%	40	40.4%	99	52.1%
Men	32	35.2%	59	59.6%	91	47.9%
Spanish nationality	90	98.9%	94	94.9%	184	96.8%
Median age (years) (P25–P75)	77.2 (70.0–83.9)		76.8 (72.4–80.8)		77.1 (71.8–82.5)	
Residence						
Residence in urban setting	39	42.9%	54	54.5%	93	48.9%
Residence in semi-rural/rural setting	52	57.1%	45	45.5%	97	51.1%
Home owner	86	94.5%	86	86.9%	172	90.5%
Median co-habitants (*n*) (P25–P75)	2 (1–2)		2 (1.5–2)		2 (1–2)	
Education						
Primary education—none or incomplete	30	33.0%	24	24.2%	54	28.4%
Primary education	38	41.8%	52	52.5%	90	47.4%
Secondary education: High school/Initial vocational training	17	18.7%	20	20.2%	37	19.5%
University education/Advanced vocational training	6	6.6%	3	3.0%	9	4.7%
Finance						
Median per capita monthly income (€)—(P25–P75)	800 (700–1100)		1000 (664.50–1350)		850 (700–1300)	
Median household monthly income (€)—(P25–P75)	1500 (1000–2100)		1600 (1200–2000)		1557.50 (1100–2000)	
Health						
Median BMI (kg/m^2^)—(P25–P75)	28.9 (25.7–31.2)		28.1 (25.5–30.5)		28.4 (25.6–30.8)	
Median monthly healthcare spending (€)—(P25–P75)	10 (0–30)		0 (0–16.5)		4.5 (0–23.75)	
Median daily consumption of medicaments (*n*) (P25–P75)	4 (2–7)		4 (3–6)		4 (2–6)	

**Table 2 nutrients-17-01717-t002:** Variables associated with increased BMI. Multivariable logistic regression model.

Variable	Odds Ratio (OR)	95% CI (Lower–Upper)	*p*-Value
SCORE VEGETABLES	0.887	0.790–0.997	0.045
SCORE CONFECTIONERY	0.923	0.843–0.999	0.049
SCORE PROCESSED MEATS	0.905	0.843–0.999	0.037

Adjusted R-squared: 0.2698.

## Data Availability

The original contributions presented in this study are included in the article. Further inquiries can be directed to the corresponding authors.

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
