# Peer review of "Association Between the Healthy Eating Index and the Body Mass Index of Older Adults: An Analysis of Food Frequency and Preferences"

_nutrients, 2025, doi:10.3390/nu17101717_

Round 1
Reviewer 1 Report
Comments and Suggestions for Authors
I have read this paper with interest. The authors report on the associations between healthy eating index and BMI in 190 older adults (65 y onwards) and hereby largely confirm previous findings and the expected outcomes. In this way, I assume that this analysis is at best confirmatory. I some way, this results in a priority decision, but that’s an editorial responsibility.
I do have reflections on the study methodology, and its reporting
To the best of my understanding, this study design can only explore association, not necessary causation. In this way, I recommend to rephrase ‘predictor’ as this contains causal language, or line 260 affect.
Along the same line, how confident are the authors that indeed lower income is the indicator, or is this rather a reflection of the education (you have collected such information, cf table 1). How has finance been analysed ? was this ‘only’ based on the pensions, or where also other revenues (renting, return on capitals) included ?
I understand the intention to collect data on 200 elderly subjects, but how confident are the authors that their cohort reflect the population on eg case mix, age, gender, other variables ? Can the authors add information on this aspect (cohort to population). This does not mean that the analysis as made is not reliable, but I would bring additional context to how to extrapolate the reported findings.
Have you considered to validate the questionnaire, like on content, face validity or similar, pilot testing ?
There is likely value to add the questionnaire as a supplement, although I understand that block 1 and block 2 are based on specific references (17, 18).
Food questionnaires are notorious for suboptimal reporting, so can the authors somewhat better reflect on this ? I understood that block 1 and block 2 were based on ‘recall’, not on prospective collection of data, or observations.
Have you considered to integrate the ‘medical history’ in your analysis, like eg blood pressure, cardiovascular events, DM treatment or others ?
Table 2: typo ‘procesed’, please check.
Author Response
To the best of my understanding, this study design can only explore association, not necessary causation. In this way, I recommend to rephrase ‘predictor’ as this contains causal language, or line 260 affect.
Thank you very much for your valuable observation. We agree that the cross-sectional design of our study does not allow for causal inference. In response to your suggestion, we have revised the manuscript by replacing the term "predictor" with "associated factor" throughout the text. Specifically, in lines 43, 49 and 160, we have rephrased the sentence to avoid implying causality. We believe these changes better reflect the observational nature of our findings.
Along the same line, how confident are the authors that indeed lower income is the indicator, or is this rather a reflection of the education (you have collected such information, cf table 1). How has finance been analysed ? was this ‘only’ based on the pensions, or where also other revenues (renting, return on capitals) included ?
Thank you very much for your thoughtful comment. We appreciate your suggestion regarding the potential confounding effect of education level on the association between income and dietary quality. In our study, we collected information on both individual monthly income and total household income, including the contributions of all cohabitants. Participants were asked to report their total regular income from all sources, not limited to pensions. This included other revenues such as rental income, investment returns, and other financial assets when applicable.
Although education level was also collected (as shown in Table 1), we did not include both variables simultaneously in the multivariable model due to concerns about multicollinearity, given the conceptual and statistical overlap between income and education. To address this, we discuss in the manuscript that both variables are likely to capture overlapping dimensions of socioeconomic status, and that their individual effects cannot be fully disentangled in this study. We have clarified this limitation in the Discussion section.
I understand the intention to collect data on 200 elderly subjects, but how confident are the authors that their cohort reflect the population on eg case mix, age, gender, other variables ? Can the authors add information on this aspect (cohort to population). This does not mean that the analysis as made is not reliable, but I would bring additional context to how to extrapolate the reported findings.
Thank you for your insightful comment. We agree that representativeness is key for interpreting the generalizability of findings. Although the study did not use population-wide probabilistic sampling, participants were randomly selected from among eligible individuals attending primary care centers in both urban and rural areas of Andalusia and Catalonia. This approach ensured diversity in terms of age, gender, and socioeconomic background. We have clarified this point in the Methods section to better contextualize the extrapolation of results.
Have you considered to validate the questionnaire, like on content, face validity or similar, pilot testing ?
Thank you very much for your insightful comment. Indeed, we conducted a pilot study with ten participants to assess the clarity, relevance, and comprehensiveness of the questionnaire items (face and content validity). Based on the feedback obtained, minor adjustments were made to the wording and sequencing of questions. This process helped ensure that the questionnaire was appropriate for the target population. We have now added this information to the Methods section to clarify the validation process.
There is likely value to add the questionnaire as a supplement, although I understand that block 1 and block 2 are based on specific references (17, 18).
Thank you for your suggestion. The complete questionnaire, originally administered in Spanish, is available from the corresponding author upon reasonable request. If required, an English translation can also be provided.
Food questionnaires are notorious for suboptimal reporting, so can the authors somewhat better reflect on this ? I understood that block 1 and block 2 were based on ‘recall’, not on prospective collection of data, or observations.
Thank you for this important observation. We acknowledge that food questionnaires based on self-report and recall, such as the instruments used in Blocks 1 and 2, are subject to potential biases, including underreporting or overreporting of intake. We have now added a statement in the limitations section to better reflect this issue. (line 301)
Have you considered to integrate the ‘medical history’ in your analysis, like eg blood pressure, cardiovascular events, DM treatment or others ?
Thank you for your insightful suggestion. Indeed, information regarding participants’ medical history—including variables such as blood pressure, cardiovascular events, diabetes treatment, and other relevant clinical data—was collected as part of the survey. However, the present analysis focused on dietary patterns and sociodemographic factors. We plan to address the relationship between medical history and nutritional status in a subsequent, dedicated analysis.
Table 2: typo ‘procesed’, please check.
Thank you for noticing this typographical error. We have corrected “procesed” to “processed” in Table 2.

Reviewer 2 Report
Comments and Suggestions for Authors
The study’s results are interesting. Some modifications may increase the quality of the paper.
- Why is the IASE thought to be suitable for this study and elderly population among some questionnaires of diet quality and pattern?
- Age-stratified nutritional approaches are required recently. Are the results of the IASE in the study useful to produce the age-stratified diets?
- Super-elderly would be a specific population for diet approach because of easy development of sarcopenia and malnutrition by poor diets. Age-stratified nutritional approaches are necessary; thus, for super-elderly, age-stratified sub-analyses might produce the relevant evidence.
- In Methods, inclusion and excluded criteria should be stated concretely.
- The validation of IASE could be discussed more with any references.
- In Abstract (line 33), what is IASE?; the definition can be added, or the abbreviation is not needed (because the abbreviation was not used in other parts. What is the abbreviation of IASE? Is it able to be fully spelled out?
- In Abstract (line 49), is the expression of ‘obesity’ changed to ‘overweight/obesity’?
- In Abstract (line 50), what are considered as the type and content of education that is effective for diets in the elderly? Elderly patients are prone to have sarcopenia, and the education is able to be specified..
- In Methods (line 126), how was concretely the calculation performed?
- In Results (line 213), what is SCORE IASE?; we can see the sudden appearance of SCORE index. Please explain or introduce it more.
- In Results (line 221), the rho value was low. Is the correlation really significant?
- The paper should have a full paragraph of study limitations.
- If infirmed consent was given in a ‘written’ type, the word ‘written’ should be added in line 317.
- In line 155, the end of sentence needs to have the period ‘.’.
Author Response
- Why is the IASE thought to be suitable for this study and elderly population among some questionnaires of diet quality and pattern?
Thank you for your question regarding the suitability of the IASE. The Healthy Eating Index for the Spanish Population (IASE) was selected because it is specifically designed and validated for use in the Spanish population, aligning with both the Mediterranean diet and national dietary guidelines. The IASE has been previously used in studies assessing diet quality in various Spanish populations, including individuals over 65 years of age, demonstrating its applicability and relevance for older adults. Its structure and scoring system are simple, culturally adapted, and practical for use in community-based studies. This makes the IASE an appropriate and effective tool for evaluating diet quality in our study. We have clarified in the Methods section the reasons for choosing the IASE and have included references to previous studies in which the index has been applied to older adult populations.
- Age-stratified nutritional approaches are required recently. Are the results of the IASE in the study useful to produce the age-stratified diets?
Thank you for your comment. The IASE measures overall diet quality but does not directly provide age-stratified dietary recommendations. Our study is focused on a relatively older population, in which dietary patterns are likely influenced more by individual health status than by chronological age. While our findings may help inform the development of tailored nutritional guidelines, capturing meaningful age-stratified differences within this age range would require a substantially larger sample size to achieve adequate statistical power. Such an analysis falls beyond the scope of the current study but represents a relevant avenue for future research.
- Super-elderly would be a specific population for diet approach because of easy development of sarcopenia and malnutrition by poor diets. Age-stratified nutritional approaches are necessary; thus, for super-elderly, age-stratified sub-analyses might produce the relevant evidence.
Thank you for your valuable comment. We agree that the “super-elderly” represent a distinct subgroup with unique nutritional needs and risks. While our sample size limited our ability to conduct age-stratified sub-analyses in this study, we recognize the importance of this approach and plan to address it in future research with larger samples.
- In Methods, inclusion and excluded criteria should be stated concretely.
Thank you for your observation. We have now specified the inclusion and exclusion criteria more concretely in the Methods section.
- The validation of IASE could be discussed more with any references.
Thank you for your suggestion. We have expanded the discussion of the validation of the IASE in the Methods section and have added a relevant reference to support its validity.
- In Abstract (line 33), what is IASE?; the definition can be added, or the abbreviation is not needed (because the abbreviation was not used in other parts. What is the abbreviation of IASE? Is it able to be fully spelled out?
Thank you for your observation. We have now defined the abbreviation “IASE” (Healthy Eating Index for the Spanish Population) the first time it appears in the Abstract and have ensured consistent usage throughout the manuscript.
- In Abstract (line 49), is the expression of ‘obesity’ changed to ‘overweight/obesity’?
Thank you for your careful reading. We have revised the Abstract to consistently use the term “overweight/obesity” instead of just “obesity” where appropriate, to better reflect the scope of our findings.
- In Abstract (line 50), what are considered as the type and content of education that is effective for diets in the elderly? Elderly patients are prone to have sarcopenia, and the education is able to be specified..
Thank you for your comment. We agree that more specific information about the type and content of effective nutrition education is important. In the revised Abstract, we now specify that targeted nutrition education for older adults should include guidance on balanced diets, adequate protein intake, and strategies to prevent sarcopenia.
- In Methods (line 126), how was concretely the calculation performed?
Thank you for your comment. We have now provided a more detailed explanation of how the sample size was calculated in the Methods section.
- In Results (line 213), what is SCORE IASE?; we can see the sudden appearance of SCORE index. Please explain or introduce it more.
Thank you for your comment. We have now introduced and explained the SCORE IASE in the Methods section, clarifying that it refers to the numerical score obtained from the Healthy Eating Index for the Spanish Population, which reflects adherence to healthy dietary patterns.
- In Results (line 221), the rho value was low. Is the correlation really significant?
Thank you for your observation. We agree that the correlation coefficient (rho) is low, indicating only a weak association despite reaching statistical significance. We have now clarified this point in the Results emphasizing the limited strength of the correlation and the need for cautious interpretation
- The paper should have a full paragraph of study limitations.
Thank you for your valuable suggestion. We have now added a full paragraph on the study limitations,
- If informed consent was given in a ‘written’ type, the word ‘written’ should be added in line 317.
Thank you for your comment. We have now specified that informed consent was obtained in writing and have added the word “written” in line 317.
- In line 155, the end of sentence needs to have the period ‘.’.
Thank you for your observation. We have added the missing period at the end of the sentence in line 155.

Round 2
Reviewer 1 Report
Comments and Suggestions for Authors
the authors have revised their paper along the suggestions provided, not additional comments or concerns
Author Response
Thank you very much for reviewing our manuscript. We greatly appreciate your valuable suggestions, which have significantly improved our paper.
Reviewer 2 Report
Comments and Suggestions for Authors
The paper was improved. The reference 16 could be changed to more accessible references to all readers, which are written in English.
Author Response
The paper was improved. The reference 16 could be changed to more accessible references to all readers, which are written in English.
We thank the reviewer for the constructive feedback, which has been very helpful in improving the manuscript. Regarding the suggestion to replace Reference 16 with English-language sources, we acknowledge that the language may limit accessibility. However, we have chosen to retain this reference because it provides the most complete account of the Eating Matters project, including methodological details and results not available elsewhere. We believe its inclusion is important for accuracy and proper citation of the original work. We also welcome any future opportunity to publish further results of the project in English.